# Complex Network-Based Research on the Resilience of Rural Settlements in Sanshui Watershed

**Jizhe Zhou**  **and Quanhua Hou ***

School of Architecture, Chang'an University, Xi'an 710061, China; 210075@chd.edu.cn
* Correspondence: houquanhua@chd.edu.cn

**Abstract:** In the context of farmland afforestation and urbanization, it is necessary for the small watershed rural settlements in the hilly–gully Loess Plateau to coordinate spatiotemporal changes and take the path of resilience development. In the case of the Sanshui Watershed, this paper investigates the rural settlement systems based on complex networks, and develops a research framework of "spatial simulation–resilience evaluation–spatial planning". The results include the evolution trends of settlement space from present to future, as well as its spatial resilience in static and dynamic states. In this study, a total of six central villages and six types of rural development are finalized, and the study area possesses a prolonged spatiotemporal resilience when 29 villages remain, thus forming an ideal spatial pattern of "rural corridor zones + characteristic towns". The findings of this study can represent guidance for resilience development in small watershed villages and provide a basis for guiding the regional urban–rural integration, village layout, as well as resource allocation and construction.

**Keywords:** small watershed; complex network; spatial resilience; resilience assessment; resilient planning

## 1. Introduction

The hilly–gully Loess Plateau is a key area for farmland afforestation in northwest China, which is also a core area of fast urbanization. The local environment is full of dense gullies, the terrain is undulating and broken, the ecological environment is fragile, and the soil erosion is particularly serious [1]. Over a long period of time, the unfavorable natural environment has caused the backward movement of rural economic and social development, and some residents still live in poverty. The Grain for Green project (GGP) was implemented in the Loess Plateau in 1999, and vegetation and ecological environment has been significantly improved since about 2006 [2]. However, the reduction in farmland seriously affects the agricultural quality and farmers' livelihoods. Meanwhile, most of the rural population move into cities due to there being better job opportunities, which led to the increasingly prominent problems of rural hollowing, labor shortage, low agricultural efficiency, unreasonable resource utilization, public service mismatch, etc. [3]. Despite the inevitable decrease in rural settlements in these areas, they will not become extinct, nor should they decay. It is a necessity to seek a resilient development path that features continuous adaptation, immediate transformation and innovativeness [4]. The hilly–gully Loess Plateau consists of numerous small watershed units, which are the basic geographical units segmented by natural divides [1]. The rural settlements within the watershed, which are connected closely and affected by the regional central cities, manifest integrity, complexity and dynamics. Among them, the integrity is reflected in the fact that the small watersheds are the cradle of settlement generation and development, where relatively closed settlement clusters have been formed during historical evolution influenced by the relationships among the clan system, production, politics, defense, etc. [5]. Meanwhile, the rural settlements within the watershed still maintain certain population movements such as labor migration and amenity migration with the surrounding regions/cities, which together form a closely connected whole unity [6]. The complexity is reflected in the

diverse directions and complex features of population and production flows. Moreover, new features will emerge after the differentiation and reorganization of various factors [7]. As for the dynamics, they are reflected in the continuous passive adaptation of rural settlements under the influence of farmland afforestation and urbanization [1]. Accordingly, small watershed villages and surrounding regions/cities form a complex system, which entails systematic thinking in the resilience exploration for settlements therein. Spatial resilience is the contribution of spatial attributes to the feedbacks that generate resilience in ecosystems and other complex systems. To operationalize and quantify it, consideration of asymmetries, connectivity and information processing is warranted [8]. Allen argues that complex network theory is useful in the research of spatial resilience, since it illustrates how the spatial resilience can be influenced by the position of a system (e.g., a wetland or a city) and its connectivity within a network of similar systems [9]. Sanshui Watershed is one of hundreds of small watersheds on the hilly–gully Loess Plateau. The core of sustainable village development in a small watershed lies in how to arrange these villages to efficiently guide the cities, and to ensure that the process reflects resilience instead of blind migration. The purpose of this research is to provide a resilient planning method based on system thinking, which could reveal the resilience mechanism of small watershed rural settlements under spatiotemporal changes, and to propose rural resilient planning schemes and corresponding development proposals. It covers the related techniques of spatial simulation, resilience evaluation and spatial planning.

## 2. Literature Review

Resilience and resilient planning of rural settlements, as the frontier of sustainability research, can be traced back to the adaptive management proposed by Holling, which is a concept used for analyzing the stability and resilience of a system under uncertain disturbances and for putting forward corresponding adaptive management solutions. There are two major types of methods for quantitatively studying spatial resilience. The first type is the resilience index-based evaluation methods represented by the standard analytical framework from the Resilience Alliance, which are methodologies based on the spatial element superposition that covers indices like resources, economy, use rights and public facilities [10,11]. However, rural settlements within the watersheds are a complex adaptive system, where the sum of elements is not equal to the whole, and generation of new attributes is possible through interconnection between elements. The second type is the relationship network-based research methods. Folke et al. claimed that the resilience of rural settlements can be reflected by the self-organizing capability of social networks for teams and participating organizations [12]. Related studies covered safety and disaster prevention [13], self-organizing capability [14], sustainable agriculture [15], resilience contraction under urban influence [16], etc. in rural settlements. The elements, connections and indicators in the aforementioned studies mostly lack spatial feedback, so spatial planning could not be achieved. Spatially, according to Masnavi et al., the form, layout and attributes of urban spatial organization could reveal spatial resilience. In addition, it could be quantitatively studied by such indicators as diversity, redundancy, coherence and efficiency [17]. Janssen argued that a system had spatial network features, and thus its resilience could be revealed by the spatial network resilience [18]. Relevant research covered the resilience of rural ecological space based on an ecological network, the village system optimization based on social networks, and the spatial resilience based on social–ecological networks [19–21]. Although such studies enable spatial planning after spatial network-based simulation and evaluation, they mostly lack temporal dynamic variations. Additionally, there are rather scarce measures and indicators of resilience evaluation that are based on relationship networks.

Complex networks are an important means of investigating complex systems, which are based on graph theory. They could reveal the mechanism of complex system operation by simulating their agents and connections through nodes and edges. For a complex system, its evolution mode can be obtained by the diffusion pattern of a complex network;

its spatiotemporal properties can be uncovered by the network architecture and features, and its steady state can be derived by assessing the network robustness and invulnerability [22]. Currently, complex network theory has been applied for the spatial simulation and resilience evaluation of urban systems. In spatial simulation, urban spatial dynamics could not only be modeled based on spatial topology networks such as roads, subways and infrastructure, but also be simulated by hybrid networks based on infrastructure network composition with citizens, housing, service facilities, etc. [17,23]. In resilience assessment, measurement formulas and indicators are introduced based on the network, such as independence, collaboration, redundancy, efficiency, and stability [17,24]. Thus, the continuous simulation space system changes and the abundant resilience evaluation methods make it possible for complex network theory to be used in the resilience research of rural settlement systems. Network perspective research on rural settlement systems reflects the spatial structure of settlements via the network architecture, which often takes administrative villages as the nodes, and inter-village social ties, characteristics and quantity as the edges [20]. However, relevant studies are generally based on questionnaire data, which restricts the scope, time interval and quantity of surveys, so the system complexity and dynamics can barely be expressed, thus meaning that the quantitative results lack accuracy. Meanwhile, complex network-based methods not only allow spatiotemporal evolution simulation of rural settlements and offer diversified indicators for resilience evaluation but can also integrate and display large and small multivariate heterogeneous data in the form of "graphs". Therefore, this could effectively make up for the shortcomings of the existing research.

## 3. Methods

Taking the rural settlements in the Sanshui Watershed as an entire research object, this study constructs a complex network by utilizing large-sample, full-coverage, spatiotemporally continuous vectored cellphone signaling big data as the basis for establishing social connections and using small data such as rural questionnaire interviews as the basis for network attributes and examining connections. Based on this, a research framework of "spatial simulation–resilience evaluation–spatial planning" could be developed. Present-to-future spatiotemporal changes of rural settlements are simulated based on the structure of the complex network and its variations, and the static and dynamic resilience of rural settlements during their evolution is quantitatively evaluated. Finally, details of resilient planning for rural settlement space are decided.

### 3.1. Data and Preprocessing

#### 3.1.1. Research Data

Villages in the Sanshui Watershed, the object of this study, are located within Xunyi County in the hilly–gully Loess Plateau, which is under the jurisdiction of Xianyang and have certain connections with Xi'an. It covers an area of 1203.3 square kilometers. Research data comprise cellphone signaling data, questionnaire interview data and map vector data. Among them, the cellphone signaling data were purchased from Intelligent Footprint company, and come from the active and passive records of interactions between the users' cellphones and the base stations, including such behaviors as powering on and off, making calls, sending and receiving SMSs, surfing the Internet, periodic location updating, location area switching and 3G-to-4G conversion. The above behaviors will be recorded based on the time, frequency, distance, orientation between the cellphone user and the base station, and become the basis for determining the location of the user. Separately in October 2018 and December 2019, recordings were performed each for 14 days, with an average daily record number of approximately 20,000. In addition, 20 questionnaires were distributed to each administrative village in the study area, which involve population, per capita income, arable land area, major industry, landscape resources, historical resources, public service facilities, disaster areas, farmland afforestation zones, etc. The map vector data consist of 1:1000 topography, as well as regional and rural boundaries.

### 3.1.2. Data Preprocessing

Currently, administrative villages are the basic and minimum unit of rural planning and policy implementation in China. This study statistically analyzes the relational features between settlements and their nature in units of administrative villages.

(1)  Natures of settlements

Based on the existing conditions, the development potential of rural settlements covers population size, per capita income, arable land area, resource conditions, public service facilities, etc. Nature of settlements embodies the foremost function of villages in the study area, and among such functions are integrated business, culture and tourism, integrated services and agricultural production [25]. The settlement nature is determined primarily by qualitative analysis with reference to the development potential index of each settlement. Firstly, the data of population, per capita income, per arable land area, etc. should be converted into a raster (with values) in ArcGIS platform. Among the data, the levels of public facilities and historical cultural resources could be judged through their quantity and quality subjectively. Meanwhile, the levels of business and disaster impact could be obtained through the superposition of buffer zones based on the influence strength of the origins. Then, the weighted superposition all the grids (with values) based on the weight could be determined by the analytical hierarchy process. From that, the potential of rural settlements could be obtained, and nature could be identified.

(2)  Features of settlement connections

Social connections among rural settlements cover aspects such as consanguineous, geographical, working, recreational and servicing ones. Relevant research has shown that villagers' travel behavior is the basis for all kinds of social connections, and the spatial characteristics and evolutionary mechanisms of rural settlements can be effectively uncovered based on the direction and volume of travel [20,26]. The travel behavior can be revealed through the "flow" in cellphone information, which represents the location change of cellphone users recognized by the base station. Localization and extraction of flows is mainly accomplished by the Intelligent Footprint company, and the details have been described in our previous studies [27]. Initially, depending on the location, power, and coverage of a base station, the area is divided into multiple square "base areas" with generally 3–5 base stations inside and around it. When the user continues to contact the next base station in a base area or frequently switches back to the original base station after accessing a new base station, the user will be identified as at a "stay point" if this state remains unchanged for half an hour. The stay point is initially identified as located in the base area. Then, according to the communication frequency and distance between the user in the base area and its neighboring base stations, the location centroid algorithm is used to interpolate and calculate more accurate coordinates of the stay point. Finally, when the stay point (start point) changes in intervals of 1 h, the arrival point is recognized as the end point. If the user finishes the trip and the stay point does not change, the farthest stay point reached will be identified as the end point [7]. Features of inter-settlement social connections include preferences, trends and the volume of connections. To reflect these features and eliminate trivial, irrelevant and incidental social connections that appeared over the course of 28 days, it is necessary to further extract the "significant flows" among the connections, i.e., the flows greater than a given threshold. Achieving this requires multiple linkage analyses that are used in the transportation engineering field [20]. Initially, the flows from each node need to be arranged from large ($X_1$) to small ($X_k$). The expected

flow traffic set of nodes is $\{\hat{X}_j\}$, where $j \in E$. $E$ represents the set of settlement nodes in the network. The computational formulas are shown in (1).

$$
\begin{aligned}
&\text{Step 1}: \hat{X}_1 = \sum_{j=1}^{k} X_j, \hat{X}_2 = \hat{X}_3 = \ldots = \hat{X}_k = 0 \\
&\text{Step 2}: \hat{X}_1 = \hat{X}_2 = \frac{1}{2}\sum_{j=1}^{k} X_j, \hat{X}_3 = \hat{X}_4 = \ldots = \hat{X}_k = 0 \\
&\text{Step } i: \hat{X}_1 = \hat{X}_2 = \ldots \hat{X}_i = \frac{1}{i}\sum_{j=1}^{k} X_j, \\
&\hat{X}_{i+1} = \hat{X}_{i+2} = \ldots = \hat{X}_k = 0 \\
&\text{Step } k: \hat{X}_1 = \hat{X}_2 = \ldots \hat{X}_k = \frac{1}{k}\sum_{j=1}^{k} X_j
\end{aligned}
\tag{1}
$$

Next, the degree of fit between the expected and actual flow volumes is determined by the decision coefficient $r^2$, whose computational formula is shown in (2). If the decision coefficient $r_j^2$ in the $j$-th step is the largest, then the top $j$ flows of the node will all be significant flows. Significant flows have directions and volumes, which are present between closely connected settlements.

$$
r^2 = 1 - \frac{\sum_{j=1}^{k}\left(X_j - \hat{X}_j\right)^2}{\sum_{j=1}^{k}\left(X_j - \overline{X}\right)^2}
\tag{2}
$$

*3.2. Spatial Simulation*

3.2.1. Topological Simulation of Settlement Space

A rural settlement system is constituted jointly by the functional agents in the system and their mutual connections, which naturally form a complex network in space. In a rural settlement system, the rural settlements are the smallest spatial units that carry the subjects, connections and functions, which exist in the form of nodes in a complex network, and whose nature is the node attributes. During preliminary survey research, we found that the rural settlements in Sanshui Watershed have relatively close connections with Xunyi County, Xianyang City, and Xi'an City. Thus, there are also three city nodes (Xunyi, Xianyang and Xi'an) in the complex network. If connection preferences exist between settlements, then there will be edges between the corresponding nodes. Attributes of these edges include the direction and volume of connections.

Future evolutionary outcome of settlement space is based on the topological simulation of scenario analysis results. As one of the foremost methods for predicting the future trends of systems and achieving resilience research [28], scenario analysis generally describes the possible future situations based on suppositions while integrating some related individual forecasts into an integrative forecast, which aims to mitigate negative consequences and enhance positive effects [29]. Therefore, the results of rural planning can also be regarded as the results of scenario analysis. With reference to relevant literature, the scenario analysis-based evolution of local rural settlement space involves the trend analyses of population mobility, urban influence and relational changes [27,30–32].

The trends of urban and rural population mobility can be obtained by consulting the dominant flow analysis used in transportation engineering, i.e., through direction computation of the "maximum flow" between nodes [33]. This study selects the top social mobility regarding flow traffic among inter-rural and urban–rural settlement nodes as the dominant flows and collects the second and third largest flows for verification of research results. The urban influence trends can be estimated by consulting Wheeler's information flow-based method of calculating the control space range for U.S. metropolitan areas. A relevant computational formula is shown in (3), where $X_{l\_c}$ and $X_{l_{l'}}$, respectively, denote the 28-day volumes of connection flow between city $l$ and village $c$, and between villages $l$ and $l'$.

$$
C_l = ln\frac{X_{l\_c}}{X_{l\_l'}}
\tag{3}
$$

The analysis of relational change trends covers two categories. First are the variation trends of social relations, which determine the results of gain and loss variations of settlement nodes (Table 1). The latter category involves the interdependency variation trends of social–ecological networks, which determine the varying results of settlement nature (Figure 1) [32]. Among them, the ecological network is constructed by consulting Allen and related mature approaches, which are not detailed herein. The results are shown in the Figure A1 [9].

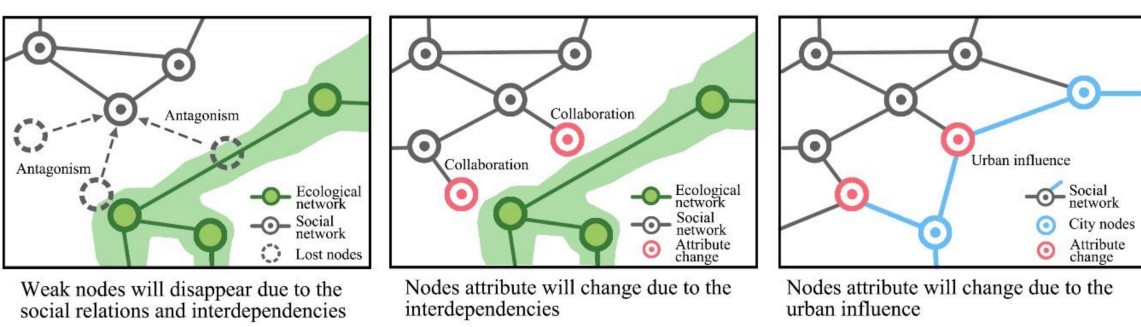

**Figure 1.** Several situations of nodes affected by interdependencies.

**Table 1.** Flow directions and meanings of significant flows.

| Description | The Significant Flow of A only Flows to B | The Significant Flows of both A and B Flow to each other | The Significant Flow of A Flows to Two or more Nodes | The Significant Flows of Two or more Nodes Flow to A |
|---|---|---|---|---|
| Spatial structure | | | | |
| Meaning | Village A has strong dependence on B | When adjacent, villages A and B are complementary or dependent. | Potential competition exists between the target villages in village A | Village A has centrality |

Finally, the node and edge information in the present and future simulations of rural settlement space in the Sanshui Watershed is imported into Pajek software for modeling and analysis. The node information includes coordinates and village numbers, and the edge information includes the quantity and direction. Then, it is visualized in the ArcGIS platform and represented the node nature.

### 3.2.2. Spatial Change Simulation of Settlements

Spatial change simulation of settlements can be implemented through changes in nodes and edges in a complex network, which covers three types of conditions: scenario analysis, random conditions and extreme conditions (Figure 2). Among them, the scenario analysis simulation deals with the possible evolution process of future settlements, which can be implemented by randomly deleting the nodes in network and the edges connected to them on the basis of the present simulation network, with the scenario analysis results as the goal. The random deletion of nodes and edges in the remaining network is continued after achieving the network structure in scenario analysis results. In this way, the possible evolution process of settlement space can be completely revealed.

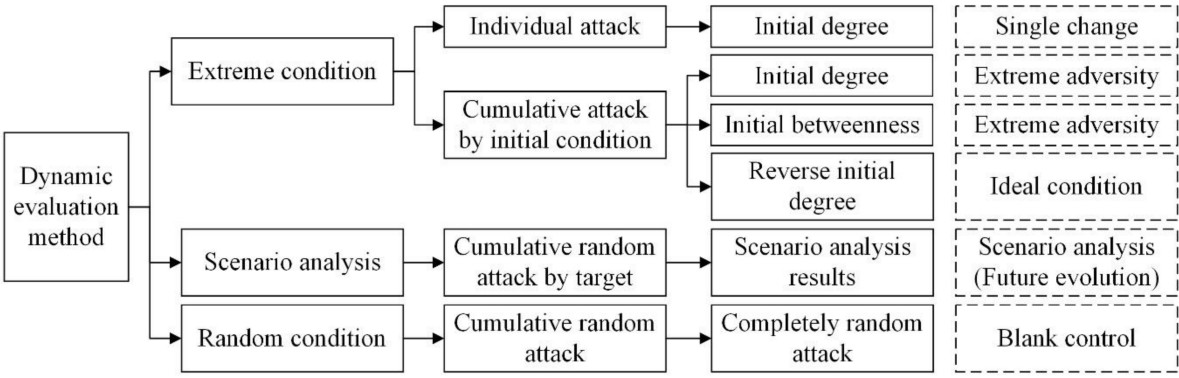

**Figure 2.** Simulation method of settlement space changes in this study.

Random condition simulation comes from the "surprises" during the evolution of complex systems [34]. When faced with unpredictable external disturbances such as planning changes, geological disasters, road construction, commercial development and policy mutations, the rural settlements will undergo random damage or relocation. This can be achieved by randomly deleting the nodes in the complex network and the edges connected to them. Analysis of random changes in the settlement space is effective in assessing the "fault tolerance" of regional settlement space, which can also be applied as a blank control for evaluating the scenario analysis results.

Extreme condition simulation involves changes that generally do not occur in the settlement space in reality, which can though appear in the ideal experiment. The spatial evolution simulation under extreme adversities (ideal conditions) can be implemented by deleting the nodes represented by settlements in the complex network and the edges connected to the nodes in order of greatest to least (from least to greatest) importance of the settlements. Through dynamic analysis of resilience variations under extreme adversities (ideal conditions), the "maximum resistance" ("optimal resistance") of the settlement space is separately evaluated. With reference to the relevant literature, the values of degree and betweenness are selected as the criteria for determining the degree of node importance [21]. Additionally, the importance degree of an individual settlement in the settlement system can be evaluated by the overall variation level of network after removing the node it represents in the network and the edges connected to the node.

### 3.3. Resilience Evaluation

With reference to the relevant literature, the indicators for evaluating spatial resilience of rural settlements include independence, collaboration, connectivity, interdependence, stability, buffering, self-organization and restorability, which are fundamentally capable of assessing the static and dynamic resilience of space systems [35–39]. The corresponding evaluation indicators for complex network resilience are node degree, structural holes, node/edge betweenness, clustering coefficients, *k*–core, core–periphery, relative size of maximum component, structure entropy and network efficiency. Figure 3 depicts the corresponding relationships.

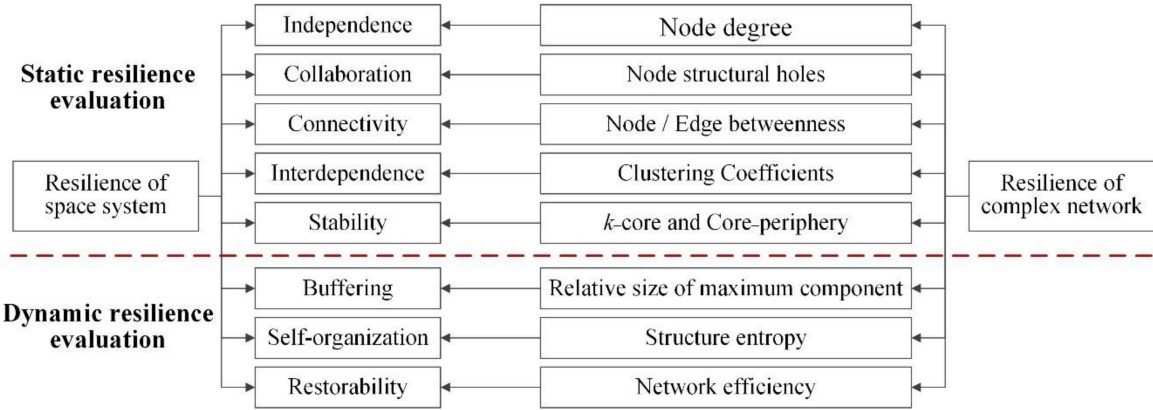

**Figure 3.** Resilience relationships between the space system and the complex network.

### 3.3.1. Static Resilience Evaluation

(1)  Independence

Independence refers to the ability to maintain the lowest acceptable level of functionality upon the disturbance of settlement systems or its subsystem in a settlement, which is also known as the self-reliance ability. The richer the local connections of a rural settlement, the stronger its independence, which can retain a certain functionality level even after losing some external connections. In a complex network, this ability can be reflected by the node degree $Ek_i$, which is the number of edges connected to the node $e_i$ in the network. As a directional indicator of dominant flows, the higher the node degree, the stronger the social connections of village it represents, the greater the competence of the village in the regional network, and accordingly the better its infrastructure and service levels. Node strength $Es_i$, which refers to the total amount of connections between edges flowing into the node, is calculated by the following formula:

$$Ek_i = \sum\nolimits_{i,j \in E, \ i \neq j} e_{ij} \tag{4}$$

$$Es_i = \sum\nolimits_{i,j \in E, \ i \neq j} we_{ij} \tag{5}$$

(2)  Collaboration

Collaboration refers to the capability of settlements to coordinate various functional agents, which is also their ability to communicate mutually and mobilize resources as a bridge. Such settlements generally have a regional competitive advantage in relative terms. In a complex network, the node structural hole $EC_i$ represents the dependence degree of node $e_i$ on other nodes, which can reflect the regional collaboration capacity of the node. For instance, when the relationships among villages in a certain area resemble a branched structure, there is often the presence of structural holes among lower-level villages, whose social ties need to be connected through an upper-level or central villages. Generally, the latter has advantages in information and resources, which occupies more structural holes as well. The smaller the structural hole value of a node, the more weakly it is constrained by the surrounding nodes, and the more options available to it. Relevant computational formulas are shown in (6)(7), where node $q$ represents the common adjacent point between nodes $i$ and $j$, and $P_{ij}$ represents the weight ratio of node $j$ among all adjacent points of node $i$.

$$EC_{ij} = \left( P_{ij} + \sum\nolimits_q P_{iq} P_{qj} \right)^2 \tag{6}$$

$$EC_i = \sum\nolimits_j EC_{ij} \tag{7}$$

(3) Connectivity

Connectivity is the ability of settlements within a settlement system to connect mutually via a network. In general, the nodes and edges at the network hub possess better connectivity. In a complex network, this ability can be reflected by node betweenness $Eb_i$ and edge betweenness $Eb_{ij}$. The greater the betweenness values, the more important the positions of nodes and edges in the entire network, which are regional hubs. Relevant computational formulas are shown in (8)(9), where $n_{st}$ denotes the number of shortest paths connecting nodes $e_s$ and $e_t$; $n_{st}(e_i)$ denotes the number of shortest paths connecting nodes $e_s$ and $e_t$ that passes through node $e_i$; and $n_{st}(e_{ij})$ is the number of shortest paths connecting nodes $e_s$ and $e_t$ that passes through edge $e_{ij}$.

$$Eb_i = \sum\nolimits_{s,t \in E, s \neq t} \frac{n_{st}(e_i)}{n_{st}} \qquad (8)$$

$$Eb_{ij} = \sum\nolimits_{s,t \in E, s \neq t} \frac{n_{st}(e_{ij})}{n_{st}} \qquad (9)$$

(4) Interdependence

Interdependence refers to the ability of a settlement in a settlement system, as part of an interconnected and integrated network, to establish functional and physical relationships with other settlements or agents and to gain support therefrom. It is generally reflected in the degree of closeness between settlements. In a complex network, this ability can be reflected by the node clustering coefficients $ECc_i$ and the network clustering coefficients $ECC_i$. The former represents the degree of node agglomeration around a node, that is, whether a regional-level cluster or community is formed. Meanwhile, the latter characterizes the overall agglomeration degree of topological network. Relevant computational formulas are shown in (10) and (11). Given the presence of substantial branched connections in the actual social network of rural settlements, the clustering coefficients of social network nodes are calculated with reference to the coefficients considering the 2-neighborhood ($CC_2$) proposed by Matthieu [40]. In the formula, $E(G_1(i))$ denotes the number of lines among vertices in the 1-neighborhood of vertex $i$, and $E(G_2(i))$ denotes the number of lines among vertices in the 1- and 2-neighborhood of vertex $i$ [41].

$$ECc_i = E(G_1(i))/E(G_2(i)) \qquad (10)$$

$$ECC_i = \frac{1}{N} * \sum\nolimits_{i \in E} ECc_i,\ 0 \leq ECC_i \leq 1 \qquad (11)$$

(5) Stability

Stability represents the ability of a settlement system to maintain steady and continuous operation, which can be manifested as the stability of the entire space system and internal subsystems. In a complex network, a higher proportion of $k$-core indicates a greater $k$ value. Accordingly, the network has more locally stable components and is more resilient as a whole, and vice versa. Core–periphery can reflect the proportion of densely networked regions in the entire network to some extent, which is also capable of finding the most resilient part of the network. The above two types of indicators jointly reflect the stability of space. Their computation is accomplished separately via the corresponding modules in Pajek and Ucinet software.

### 3.3.2. Dynamic Resilience Evaluation

(1) Buffering

Buffering refers to the ability of a settlement system to maintain its own capabilities without serious loss or degradation of its major regions or functions under external shocks. The maximum component, as the largest subnetwork in a complex network, reflects the buffering performance of system under continuous attack [42]. A relevant computational formula is shown in (12), where $KN'$ denotes the number of nodes in the maximum component, and $KN$ denotes the total number of nodes in the network.

$$KS = KN'/KN \tag{12}$$

(2) Self-organization

Self-organization is often reflected in the non-reliance on external assistance. For a system, its self-organization ability can be greatly enhanced by the diversity and redundancy of its internal functional subjects and elements [43]. In a complex network, structure entropy characterizes the disorderliness of the network system. A greater entropy value indicates more microscopic attributes of the system, which is more disorderly and possesses a stronger self-organizing capability [44]. Relevant computational formulas are shown in (13)(14), where $p(Se_i)$ denotes the probability of a node attribute being $Se_i$.

$$p(Se_i) = n(Se_i)/N \tag{13}$$

$$HSK = -\sum_{i \in K} [p(Se_i) * \text{In } p(Se_i)] \tag{14}$$

(3) Restorability

Restorability refers to the rate at which the settlement system recovers under external shocks. Having better connectivity and operational efficiency between settlements enables faster mobilization and organization of resources in the face of crises, which enhances the post-disturbance restorability of the regional rural settlements. In a complex network, network efficiency can reflect network connectivity and structural accessibility, which indirectly reflects the restorability of the network system [45]. A relevant computational formula is shown in (15), where $N$ denotes the number of nodes in the present network, and $d_{ij}$ denotes the shortest path between nodes $e_i$ and $e_j$ in the network. If the nodes are disconnected, $d_{ij}= +\infty$, and the efficiency value will be 0.

$$E(G) = \frac{1}{N(N-1)} \sum_{i,j \in E, i \neq j} \frac{1}{d_{ij}} \tag{15}$$

### 3.4. Spatial Planning

#### 3.4.1. Village–Town System Planning

On the basis of spatial simulation and resilience evaluation, villages with relative competitive advantages and development potential are selected as the regional central villages for focused construction. These central villages generally have strong regional influence and can serve the surrounding administrative villages. First of all, the nodes with higher degrees and lower structural holes are often selected as the central villages. Next, the nodes chiefly influenced by the central village nodes are identified based on the connection directions. If an influenced node has multiple edges pointing to different central villages, the node marked by the strongest edge will be regarded as the node it belongs to. Afterwards, based on the variation trends of rural development potential, villages with more prominent statuses in the future are selected as candidate central villages, which are constructed in advance. Finally, the regional village–town system is examined and improved on a full coverage basis of central village services in the regional space.

### 3.4.2. Rural Development Type Planning

To reflect the resilience and stability during rural development, the administrative villages within the watershed are classified into 6 development types, namely centralized promotion, reserved development, resilient shrinkage, urban–rural integration, characteristic protection and relocation, which is accomplished based on the results of static and dynamic resilience evaluations by consulting statutory planning requirements. Furthermore, relevant guiding directions and development guidelines are put forward.

(1) Centralized promotion type: In a complex network, they are nodes to which the primary flow trends and preferences point. They feature an exceptionally high degree and a very low structural hole index in both complex networks that are formed following the status and scenario analyses, which are regional cores of the network. Their loss affects the network structure distinctly (abrupt change), or results in a greatly lowered network efficiency (abrupt change). (2) Reserved development type: In a complex network, they are nodes to which the secondary flow preference point. These nodes possess a high degree and a low structural hole index in both complex networks that are formed following the status and scenario analyses or have an elevating status in the future. Their dynamic resilience features resemble those of the "centralized promotion type". (3) Resilient shrinkage type: These nodes are the flow preferences of a few nodes in a complex network, who will lose substantial connections in the future. They have a low degree and a high structural hole index in the present network, or will have a drastic decline in status in the future. Their loss has little or no impact on the network structure or leads to a slightly lowered or enhanced network efficiency. (4) Urban–rural integration type: These nodes are strongly influenced by urban nodes in a complex network. Spatially located on the urban fringes, they have strong connections with urban nodes, which are considerably stronger than their connections with rural nodes. Their dynamic resilience features resemble those of the "centralized promotion type" when they have strong village–village connections; otherwise, the features resemble those of the "relocation type". (5) Characteristic protection type: These nodes usually possess cultural tourism properties, or have cultural relics, preservation zones and characteristic landscapes within the villages. Their loss will result in a decrease in network structure entropy. (6) Relocation type: In a complex network, they are the starting points of mobility trends and preferences. Having considerably low degrees, they will be lost in scenario analysis, which are often structural holes. Their loss has little or no impact on the network structure, or usually leads to an enhanced network efficiency.

## 4. Results

### 4.1. Spatial Simulation Results

The results of population mobility, urban impact and relationship change trends are shown in Figure A2, and Figure 4 depicts the present and future spatial simulations of rural settlements in the Sanshui Watershed, where nodes 1-43 are rural nodes, representing 43 administrative villages in the study area. Meanwhile, nodes 44-46 stand for the city nodes with significant influences on the area. There are currently 254 significant connections (edges). In the future, 14 nodes will disappear, which will be relocated towards other villages (cities). Regional culture and tourism will be further developed, and non-agricultural nodes will increase.

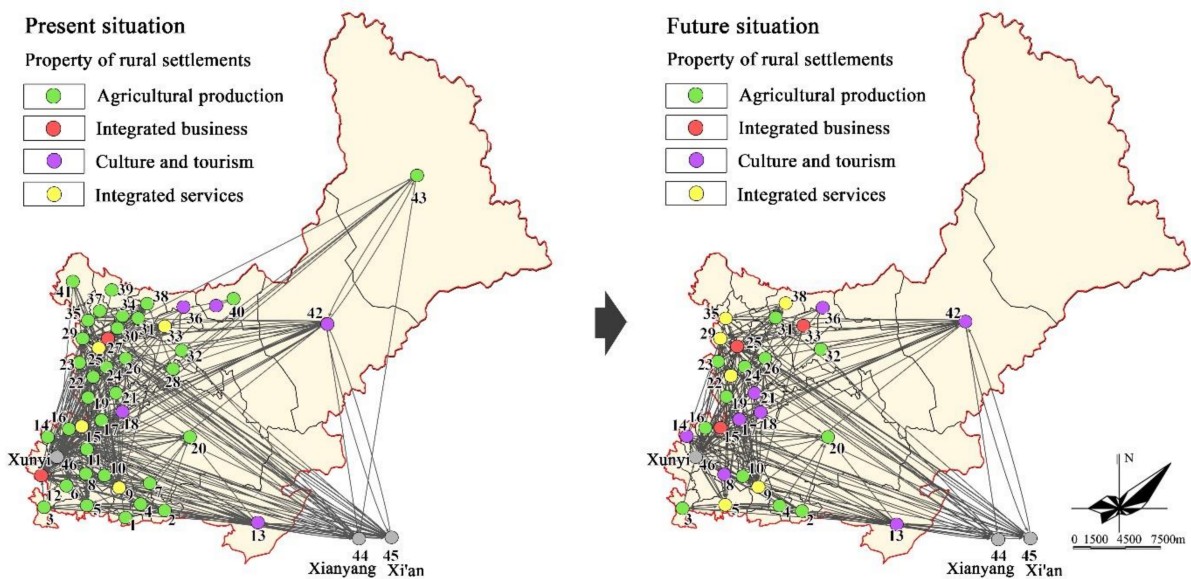

**Figure 4.** Complex network-based spatial simulation models of rural settlements in Sanshui Watershed.

### 4.2. Static Resilience Evaluations

As is clear from Figure 5, the average degree of settlement nodes is currently 6.85 for the Sanshui Watershed area, which exhibits a scale-free network feature of the complex network. The settlements in the watershed are characterized by distinct regional centralization, which gather towards the northern and southern tableland centers and the open valley region, respectively. Regional centers generally have broad development space and convenient transportation conditions. According to the evaluation of scenario analysis results, four rural settlements around the county will be merged into the cities, while the regional centralization of the northern and southern tableland regions are more distinct. The social connections within the watershed are seriously influenced by cities, with village–village connections only accounting for 51.36% of the total connections. The urban–rural connections are especially prominent for nodes 42 and 13 due to their tourism functions at the provincial and municipal levels.

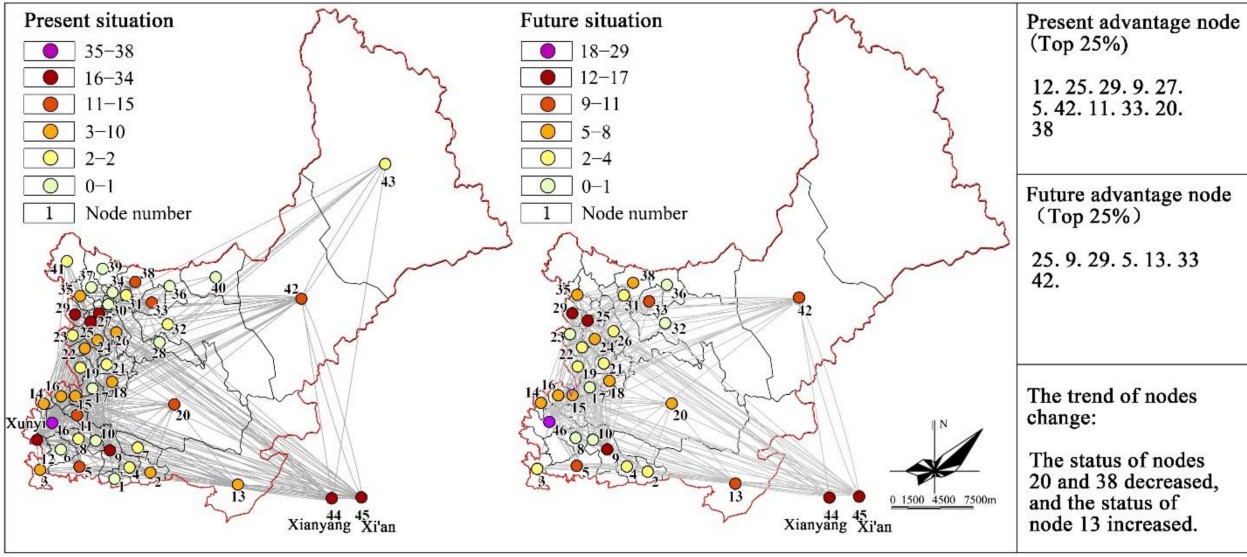

**Figure 5.** Node degrees and their changes in the complex network.

Nodes with smaller structural holes possess a better intermediary capability, which will generally be the regional centers if they also have higher degrees. As shown in Figure 6, the structural holes within the watershed are primarily distributed on the tableland margins and in the outskirts with inconvenient traffic. They are interconnected mainly through the regional centers such as nodes 12, 29, 9, 25 and 5, which establish branched social connections. Meanwhile, the regional center nodes are closely interconnected, forming a networked social connection across regions and terrains. At a threshold of 0.37, the proportion of regional structural holes is 23.26%. In scenario analysis, the proportion of structural holes in the complex network is reduced to 17.20%, thus forming more effective social connections. In the future, the regional controllability of advantageous rural settlements such as nodes 5 and 33 will be weakened, while that of node 18, which is closer to the city, will be enhanced. Moreover, the original multi-core structure of tableland regions will evolve into a single-core structure.

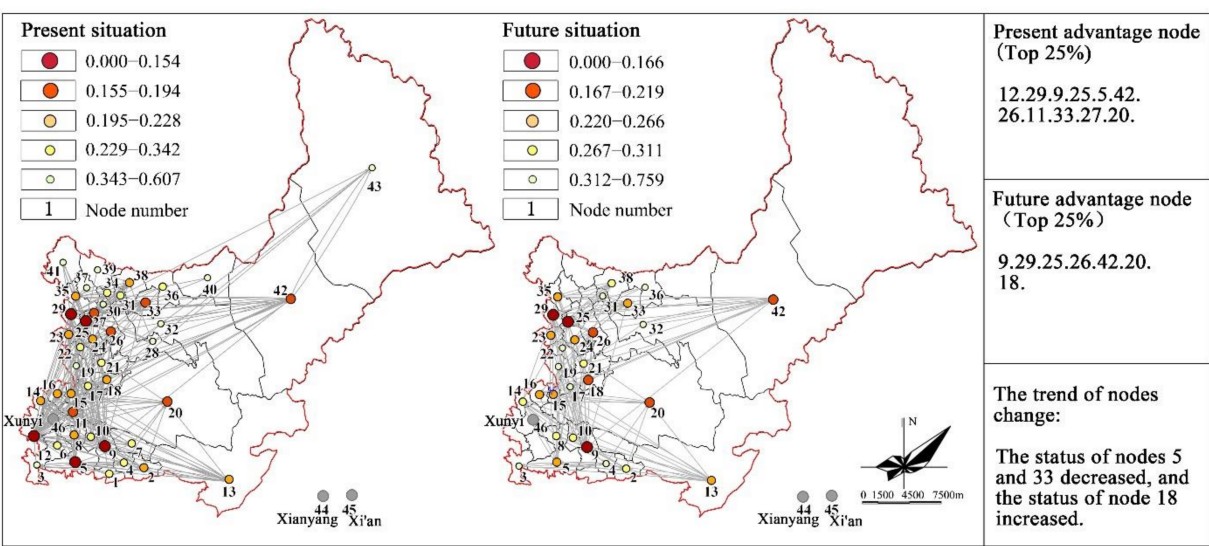

**Figure 6.** Structural holes and their changes in the complex network.

As is clear from Figure 7, more important connections within the watershed are concentrated on two branches. After passing through nodes 2 and 9 from the village node 13, one of the branches goes westwards through nodes 12, 11 and 42 to arrive at 43, while the other branch goes northwards through nodes 18, 25 and 29 to arrive at 41. In the real world, the former is the river valley axis of County Road X306–Valley, while the latter is the arterial axis of the County Road X306–Provincial Road S306–National Road G211. Based on the evaluation of scenario analysis results, the role of the river valley axis will gradually weaken, the role of the arterial axis will gradually strengthen, and a new minor axis spanning from the village node 25 through 26 to 42 will be added. Hence, rural development will be weakened by terrain factors and strengthened by traffic factors. In the future, the hub status of village nodes 2 and 13 on the southeastern fringe will be gradually replaced by node 15, which is closer to the county. Nevertheless, they will be more closely connected with large regional cities.

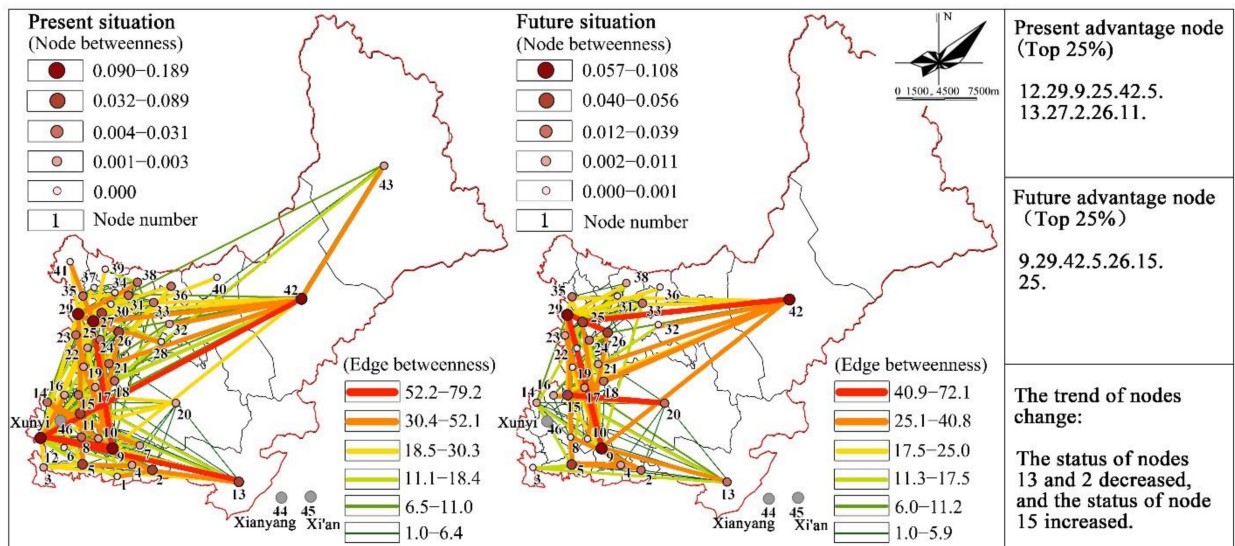

**Figure 7.** Node/edge betweenness and their changes in the complex network.

Affected by topography, the rural settlements in the watershed can barely establish large-scale agglomerations, and only form a small-scale agglomeration in the center of northern tableland around the village node 25, as shown in Figure 8. If 0.05 is used as the threshold, the nodes with lower clustering coefficients will account for 33.33%, and the overall clustering coefficient of the network will be 0.128. In the scenario analysis, the proportion of nodes with lower clustering coefficients decreases to 27.59%, and the overall clustering coefficient of the network remains at 0.128. The overall resilience of the study area fails to improve, since some core villages lose substantial social connections with surrounding villages after being inter-merged or merged into the cities. Hence, when relocating and merging the core villages, the original functions and facilities should be preserved or optimized, and the original social connections should be maintained.

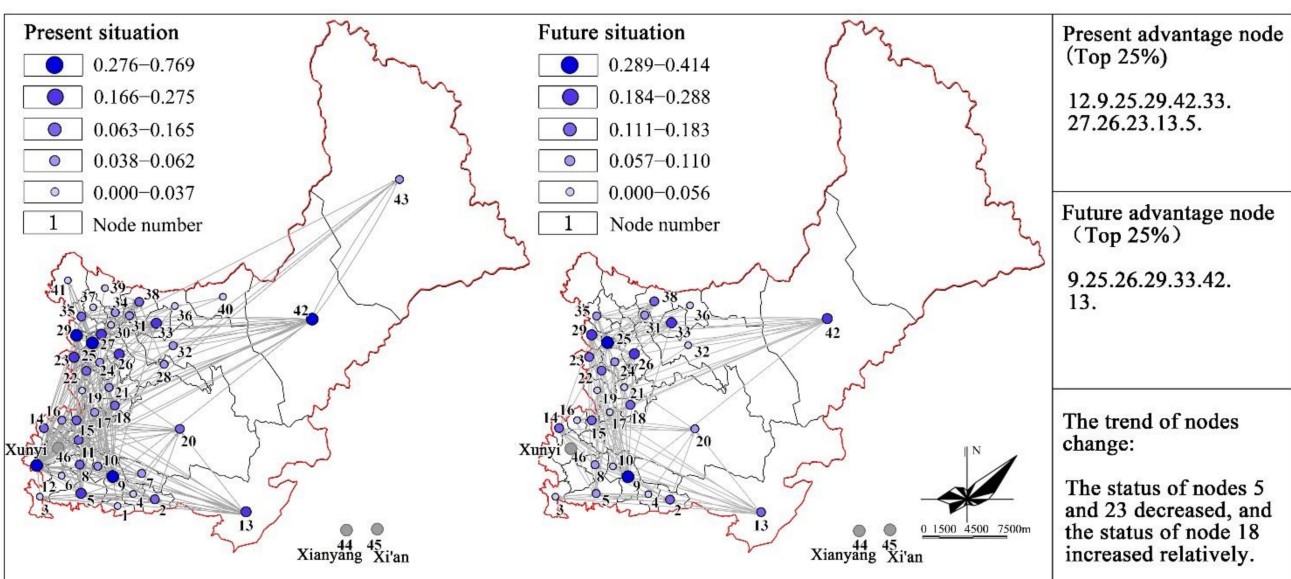

**Figure 8.** Clustering coefficients and their changes in the complex network.

From Figure 9, it can be seen that the maximum value of *k*-core in the watershed complex network is 4-core. The northern tableland center and the open valley region are relatively resilient regions. The proportion of node number in these regions accounts for 16.28%, and vulnerable regions with 1-core and below occupy as much as 25.58% in the

network. Currently, there are 20 core nodes in the network, with a proportion of 46.51%. The density of core regions is 0.429, whereas the density of peripheries is 0.039. After scenario analysis, the maximum value of *k*-core in the social network decreases to 3-core. The proportion of socially vulnerable regions increases to 27.59%, and the entire network becomes more vulnerable after inter-merging and merging of some core villages into the cities. A total of 16 core nodes are present in the network, showing an increased proportion to 55.17%. The density is 0.522 for the core regions, whereas it is 0.066 for the peripheries. Despite shrinkage of the core regions in the social network, their degree of agglomeration is enhanced. Moreover, in the future, these network cores will further gather towards the southwestern traffic arteries.

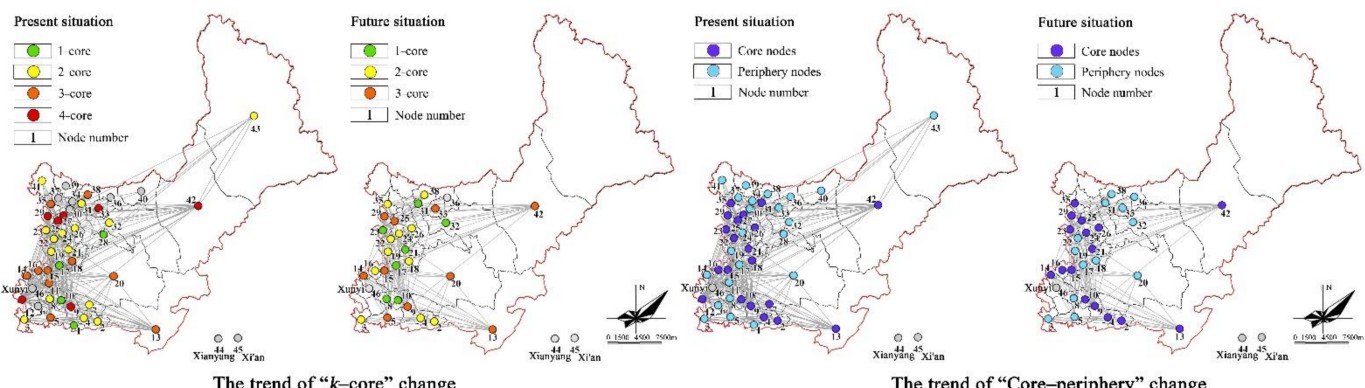

**Figure 9.** Complex network "*k*–core"/"core–periphery" indices and their changes.

### 4.3. Dynamic Resilience Evaluations

Loss of village nodes with higher degrees is often more influential to the stability of network structure, while loss of nodes with lower degrees has little or even no impact on the network. The evaluation results demonstrate that 16.28% of nodes produce a greater impact on the network after loss, such as on the village nodes 25, 29, 9, 42, 18, 2 and 26, which are the critically influential villages of resilience buffering. Meanwhile, 67.44% of nodes have little impact on the network after loss, and 16.28% of nodes have no impact on the network after loss. Among them, although the village nodes 12, 27, and 11 have relatively important statuses in the study area, their loss overall has no impact on the network due to the relatively independent components formed by them with the surrounding villages.

Structure entropy of the social network calculated based on the scenario analysis results is 1.163, revealing a 58.27% higher value than the original network. Diversified development of villages strengthens the resilience coordinating and self-organizing capabilities of the study area upon crises. In case the present complex network loses the administrative village nodes with attributes of integrated business, culture and tourism, integrated services and agricultural production, the structural entropy values of the overall network will decrease by 7.51%, 5.86%, 4.78% and -2.37%, respectively. The corresponding decreases in the case of scenario analysis are 2.51%, 1.49%, 2.11% and -0.79%. Hence, in the future, external disturbances will have drastically weakened impacts on the complex network, and the spatial resilience of rural settlements in the watershed will be enhanced by diversity and redundancy.

From Figure 10, it is clear that, in most cases, loss of nodes with higher degrees leads to a more drastic reduction in overall network efficiency in the watershed social network. Meanwhile, loss of some minor nodes can enhance the overall network efficiency, which is more conducive to the inter-settlement social connections in the network. Since such nodes account for 51.16%, over half of the administrative villages can be recommended for relocation. Under the dual influences of eco-space expansion and urbanization, the trend of resilience development for local rural settlements is precisely the large-scale shrinkage of production and living spaces. Village nodes like 12, 25, 42, 9, 29, 26, 5 and 18 have a

greater impact on the network efficiency, which are the critically influential villages of resilience restoration.

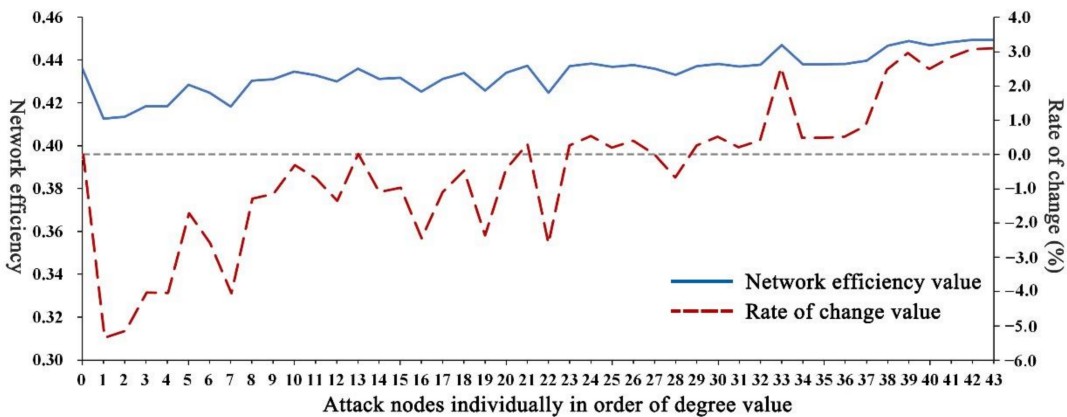

**Figure 10.** Network efficiencies and rates of change under individual attacks.

As shown in Figure 11, under extreme adverse conditions, that is, after attacking the network at the initial degree and betweenness, the numbers of removed nodes that can cause abrupt changes to the network's relative size of maximum component are both seven. Under the most ideal condition, that is, after attacking the network at a reverse initial degree, the above indicator decreases gently, and the network structure fluctuates slightly. When 14 nodes are lost to reach the goal of scenario analysis, the relative size of maximum component can be up to 96.55% of the ideal condition. It can even be equal to the ideal condition upon the 16-17th attacks, revealing good resilience buffering and resisting capabilities. It is not until further loss of 15 nodes after attaining the scenario analysis goal that the network structure undergoes abrupt changes and the indicator lags behind the blank control. Thus, scenario analysis-based resilient planning has a long validity.

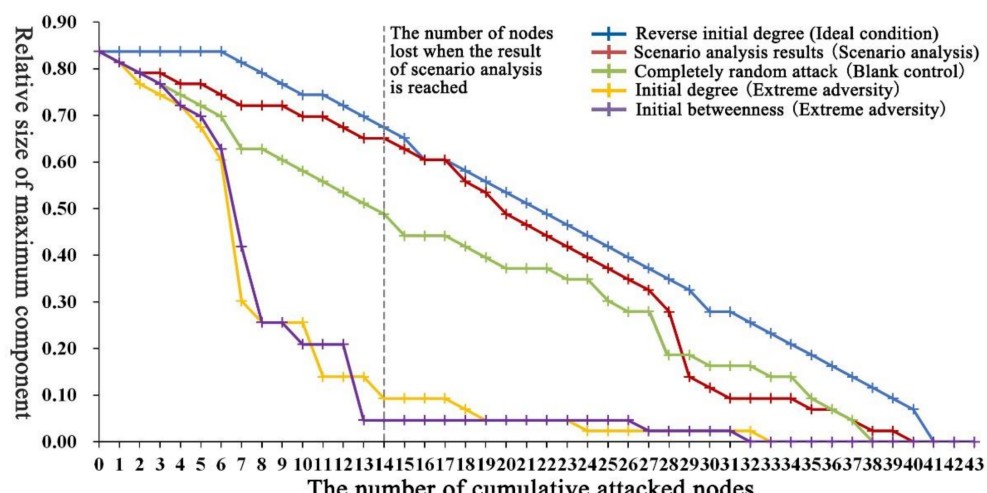

**Figure 11.** Relative sizes of the maximum component after cumulative attacks.

It is clear from Figure 12 that in terms of nature, the original rural settlements are dominated by agricultural production, with weak diversity. Thus, the changes in structure entropy of network differ obviously under five modes of attack. The settlement evolutions under both scenario analysis and ideal conditions show a period of rising structural entropy. When 14 nodes are lost, the structure entropy values under scenario analysis, completely random, initial degree and initial betweenness conditions are 99.11%, 85.51%, 69.18% and 74.78% of the ideal condition, respectively. After attaining the goal of scenario analysis, the

structural entropy values of the network are higher than the initial values. This is attributed to the removal of excessively redundant nodes with agricultural production property, and the corresponding relative increase in diversity. Afterwards, the structure entropy of the network will be kept at fairly ideal levels, so that the coordinating and self-organizing capabilities of rural settlements can continue to be maintained.

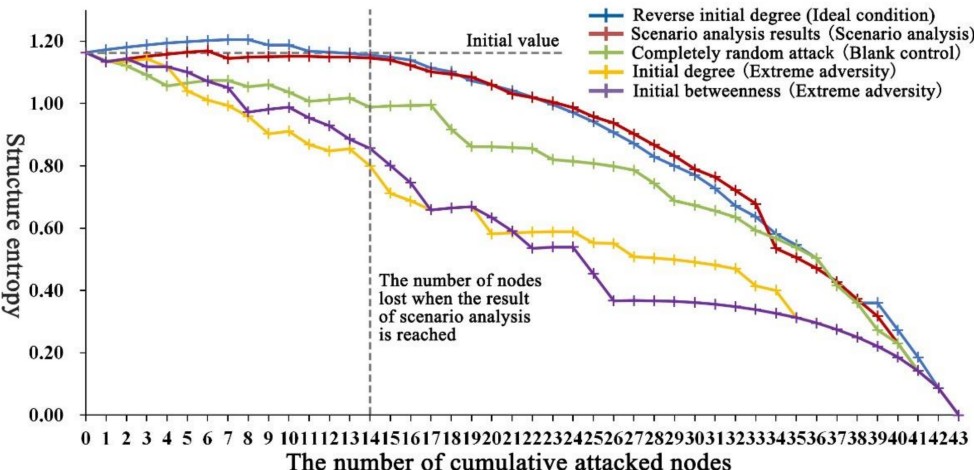

**Figure 12.** Structure entropy values of the complex network after cumulative attacks.

As shown in Figure 13, under the reverse initial degree attack, the network efficiency is always improved with the continuous shrinkage of the network from the peripheries to the centers. The settlement evolution based on scenario analysis shows an initial phase of rising network efficiency, which reaches its peak at the 16th attack, with an increase by 22.25% from the initial value. At the 28th attack, the network efficiency returns to its initial value. After fulfilling the goal of scenario analysis, the network efficiency can reach 85.87% of the ideal condition, the resilience restorability is enhanced, and higher network efficiencies can be maintained for a certain period of time, only lagging behind the blank control during the later period. In the Sanshui Watershed with complicated terrain and inconvenient communication, the adversities caused by extreme conditions are far stronger than those in the plain area (another empirical site of this study), and the restorability of watershed villages is also rather weak. Table 2 details the village nodes that are greatly influential to the regional spatial resilience in the dynamic resilience evaluation.

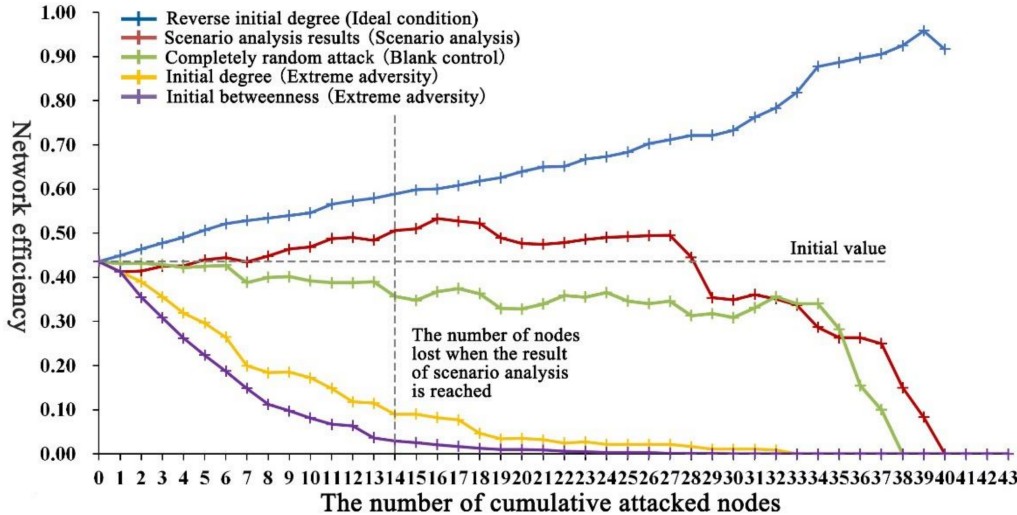

**Figure 13.** Network efficiencies of the complex network after cumulative attacks.

**Table 2.** Nodes influencing the dynamic resilience of the complex rural network.

| Evaluation Index | Reverse Initial Degree | Scenario Analysis | Completely Random | Initial Degree | Initial Betweenness |
|---|---|---|---|---|---|
| Relative size of maximum component | – | 42 | 42 | 42; 29 | 25; 13 |
| Structure entropy | – | – | – | – | – |
| Network efficiency | 20 | 29; 42; 9; 25 | 25; 12; 42; 29 | 29; 9; 42; 13 | 29; 9; 25; 42; 13 |

To compare changes in social network under the ideal condition versus scenario analysis, each six network slices are extracted from the two evolution modes for simultaneous comparison and analysis. As shown in the Figure A3, the early- and middle-phase variations based on scenario analysis differ insignificantly from the ideal condition, and the network structure distribution also exhibits a C shape around the arterial axis. During the middle and late phases, various regions still retain respective cores, which can offer convenient services to the surrounding villages. Thus, under the spatial evolution trend of regional agglomeration and road dependence, the spatial resilience of rural settlements in the watershed can maintain fairly ideal levels for a prolonged time, and the time effectiveness of scenario evolution-based resilient planning is the loss of 30 nodes.

*4.4. Spatial Planning Results*

As is clear from Figure 14, due to merging of regional core village nodes 12 and 11 into the cities, the inter-village connections between rural settlements in the urban fringes fail to reflect centrality, which only maintain the urban–village binary connections with the cities. These settlements will share public services with the cities. Based on the flow trends and preferences between village nodes, five central villages are identified within the watershed, which are generally located at the regional centers and differ greatly in scope of influence. Due to the status decline of the central village Houzhang (33) in the future, its function will be gradually replaced by Zhitian (25). Meanwhile, Shimen (13) on the southeastern side will be further independent of the original rural settlement system, which will establish a closer connection with regional large cities. It will be the future central village, which should be constructed in advance, oriented towards urban tourism. According to the urban inflow intensity, central villages 13 and 18 should be equipped with public services conforming to urban standards, such as general hospitals, vocational schools, primary and secondary schools, cultural and sports centers and nursing centers, in order to realize the equalization of public services between urban and rural areas. For other central villages, they should deploy public services in accordance with the village standards set by the local government. Based on the strength of nodes, further improvement is required to the deployment scale of public service facilities in village 25.

According to Figure 15, villages of centralized promotion and reserved development types are located mainly on both sides of arterial roads in the open tablelands; villages of urban–rural integration type are located mainly in the open valley lands and low-lying terraces on the county fringes; and villages of resilient shrinkage and relocation types are located mainly on the tableland edges and exurbs. Table 3 describes the guiding directions and development guidelines for various rural settlements. As suggested by the results of dynamic resilience evaluation, after the relocation of 14 rural settlements this time, the rural area in the Sanshui Watershed already possesses good spatiotemporal resilience, and the time effectiveness of spatial planning based on the scenario analysis results is also ideal. Such planning not only allows natural and random evolution of rural settlements following successful resilient planning, but also allows preferential selection of nodes with enhanced or slightly lowered post-attack network efficiency as the relocation villages for the next resilient planning. By 2030, there will be 29 administrative villages, about 38,900 people, and 466.7 hectares of construction land in the area. At that time,

community-based, better farmland and equal public services will be preliminarily realized. By 2040, there will be 23 administrative villages, about 2,9100 people, and 349.5 hectares of construction land in the area. Under ideal future conditions, the rural settlement space in the watershed is evolved into a spatial form of rural agglomeration along the traffic arteries that is interspersed with small towns with different characteristics.

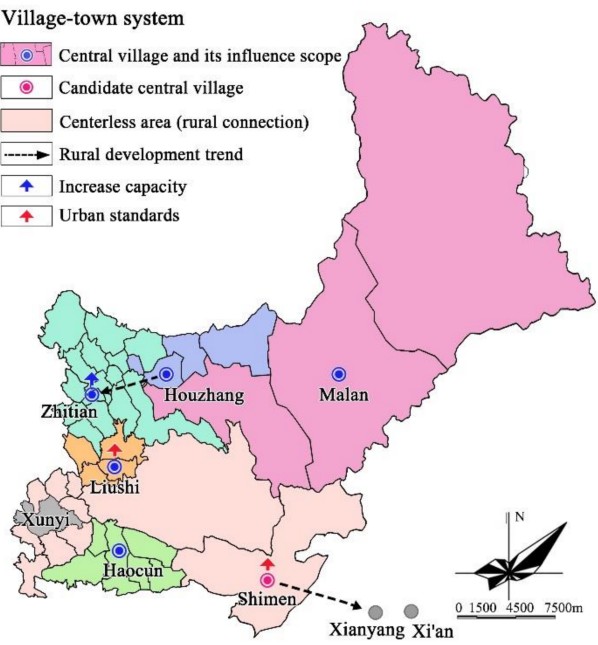

**Figure 14.** Planning results of the village-town system.

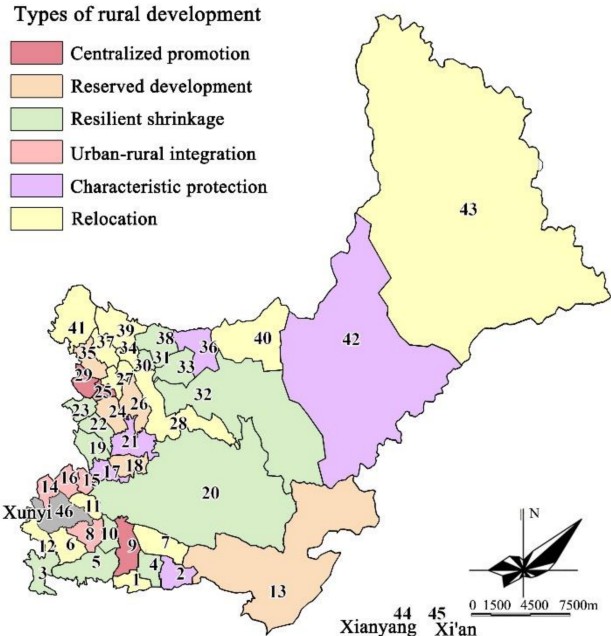

**Figure 15.** Planning results of the rural development type.

**Table 3.** Guiding directions and development guidelines for rural settlements.

| Type | Development Guidelines |
|---|---|
| Centralized promotion | Emphasize local development and optimize industrial structure; focus on the construction of new rural communities by relying on convenient traffic and guide the agglomeration of surrounding villages; beautify the village environment and optimize the public service facilities. |
| Reserved development | Emphasize local development and strengthen external connections; enhance the levels of basic public services and infrastructure; improve the rural living environment and embody the inherent style and feature of villages. |
| Resilient shrinkage | Gradually guide towards regional central villages and cities; restrict the construction lands, consolidate scattered farmlands and gradually return farmlands to forests; optimize public services and resource allocation through regional sharing. |
| Urban-rural integration | Gradually guide towards cities; encourage land circulation, retain the original functions and facilities of villages and realize urban–rural resource sharing; provide skills training to guide the urbanization of villagers. |
| Characteristic protection | Focus on local development and strengthen external connections; revitalize local culture and restore the ecological environment; reflect the iconic and unique features of rural construction. |
| Relocation | Gradually guide the outward relocation of rural settlements; villagers can farm in other villages through land replacement or can work in cities and towns after receiving subsidies for farmland afforestation. |

## 5. Discussion

### 5.1. Strengths of This Study in Rural Planning

This study comes from our team's Village–Town System planning and rural construction planning for the Xunyi County in 2020. When we adopted a traditional approach based on superposition of spatial attributes for the development potential research of various villages within the watershed, undesirable results were obtained [46,47]. For instance, Lvjia (Node 5) is a village with strengths in rural population size, arable land area, transportation conditions and public services, which is also considered to have great development potential in traditional research. In reality, however, Lvjia is a highly-hollowed declining village with a permanent population of less than 15%, since most of the villagers work in cities. In contrast, Shimen (Node 13) is a mountainous administrative village with a small rural population, where farmland afforestation has been completed, meaning that it has no arable land. Although Shimen is a disadvantaged village that should be relocated in traditional research, its permanent rural population ratio is as high as 68%, and there are also substantial amenity migrants from cities like Xi'an and Xianyang, which has an exceptionally strong external connectivity. The village develops tourism relying on its abundant surrounding landscape resources, whose actual population, income and vitality are far higher than Lvjia. The urban and rural settlements in the entire small watersheds constitute a complex system, and the above problem can only be avoided from the perspective of a relational network. Thus, in the small watershed rural areas, the social connection-based complex network reflects real spatial relationships. Network changes can better reveal the dynamic operation rules of settlement system, assess the system resilience, and achieve resilient planning.

### 5.2. Coordination between Temporal Resilience and Spatial Resilience

Spatiotemporal investigation on the resilience and resilient planning of rural settlements is a current research gap at home and abroad. As mentioned earlier, spatial resilience is the contribution of spatial attributes to the feedbacks that generate resilience in systems. Meanwhile, temporal resilience is embodied in the advocacy experimental method. The control process is precisely a course of experimental control, which emphasizes the adjustment of a control plan after learning in practice or scenario assumptions [48]. In this study, coordination between spatial and temporal resilience is preliminarily realized through the simulation and resilience evaluation of settlement space via a complex network. To be

specific, spatial resilience is embodied in the static resilience evaluation of present and future simulation networks, while temporal resilience is embodied in the resilience evaluation during five dynamic evolution processes including scenario analysis and completely randomized design, as well as in the planning proposal based on the dynamic processes. Additionally, complex network-based research also allows collaborative analysis of multivariate heterogeneous data. It can express data of different formats, types and orders of magnitudes in the form of diagrams and attributes, thereby achieving quantitative research of the settlement space resilience during evolution in mathematical language. However, this study has insufficient learning in practice, so building a dynamic feedback information platform based on network changes is required.

*5.3. Resilient Planning Proposals for Small Watershed Rural Settlements*

Rural settlements in small watersheds are a relatively closed integral system. Influenced by urbanization and farmland afforestation, the rural settlements exhibit overall features of regional agglomeration, road orientation and urban orientation. The following several aspects should be considered into the resilient planning of rural settlements: 1) The watershed periphery is generally an important zone in terms of ecological functionality. The number and scale of rural settlement patches in this area should be reduced, and the settlements should be migrated towards the open tablelands and river valleys. Furthermore, the abandoned settlement and farmland should be afforested. Nevertheless, for any region forming a closer connection with city that is independent of the closed system, its independent development can be considered. 2) Within small watersheds, the side slopes of tablelands are often ecologically-fragile and disaster-prone regions, where inefficient farmland conflicts with forest land. Historically, these regions have established many settlements for defense needs. Today, these settlements should migrate downwards to both sides of the valley roads, or upwards to the regions with convenient transportation in the tableland center. At the same time, ecological restoration should be promoted on the side slopes. 3) Roads, as the sole medium for material and information exchange with the outside world within the small watersheds, will have a more important position. It is necessary to gradually guide the roadside migration of settlements while enhancing the compactness of settlement patches, thereby further improving the regional connectivity efficiency. 4) It is recommended to control the population within each living unit of loess tableland to 10,000–15,000, in order to form relatively independent infrastructure and public service facilities. It is also necessary to expand the size of central villages, guide the agglomeration of surrounding villages, form ecological barriers at the tableland sides, and strengthen the connectivity of unit with the cities and surrounding living units.

**6. Conclusions**

In this study, the spatial resilience and resilient planning of rural settlements in small watershed is investigated by innovatively adopting the theoretical approach for complex networks. In the case of the Sanshui Watershed, the village–city linkages, which account for 48.64% of the total, show that the present rural settlements are strongly affected by the city. The rural settlements are concentrated towards three regions. Networked social connections are established among the regional central villages, while somewhat branched social connections are established between the central and surrounding villages. The development axis for the watershed is a region passed by the river valley and arterial roads, and the rural settlement clusters are formed only at the northern tableland center. The northern tableland center and the open valley land are more resilient regions, and the number of nodes in these regions accounts for 16.28%. In the future, the trend of settlement centralization will become more prominent, and the regional controllability of nodes 5 and 33 (located at the edge of the tableland) will be weakened, whose status will gradually be replaced by village 18. Furthermore, the multi-core structure of tableland regions will be transformed to a single-core structure. A minor axis will be added to the study area, the original valley axis will be weakened, while the arterial axis will be

strengthened. The status of such hub villages in the watershed as 2 and 13 (located at the scenic spot) will be gradually replaced by node 15 that is closer to the center, albeit their closer connections with large cities. In the course of urbanization, the overall centralization of settlements remains unchanged, no relatively resilient regions are present in the network, the proportion of vulnerable regions increases by 2.01%, and the network cores gradually gather towards the southwestern traffic arteries. During settlement evolution, 16.28% of rural settlements pose a great influence on the overall situation, and 51.16% of villages are recommended for relocation. Scenario analysis reveals that the resilience buffering, self-organizing and restoring capabilities can be up to 96.55%, 99.11% and 85.87% of the ideal condition, respectively. Based on resilience evaluation results, six central villages and six rural development types are identified, and the resilient planning cycle is one round. The study area achieves a prolonged spatiotemporal resilience when 29 villages remain, which forms an ideal spatial pattern of "rural corridor zones + characteristic towns". This study could predict the evolution of villages in small watersheds and evaluate their resilience. The results could arrange suburban villages to move to the cities reasonably, shrewdness and efficiency. During this process, transportation plays a leading role and gathers the rural settlements to the center of the tableland or cities. Within a certain threshold, regional service efficiency and competitiveness can be improved by removing some vulnerable villages. However, the resilience of the rural region will reduce seriously when losing some advantageous villages. In particular, the villages incorporated into the city should still maintain their original social connections. Therefore, calculating the threshold and determining the position of villages will be the primary task of resilient planning. In Sanshui Watershed, a village with the characters of the district center, industrial advantage, convenient transportation and intermediary between districts could often determine spatial resilience. However, the subcenter or high quantity may contribute less. Additionally, the villages outside the rural settlement system but with close ties with the cities should be found and their urban service functions highlighted.

By regarding rural settlements within the watershed as an integral system, this study utilizes the topological network to simulate the present-to-future spatiotemporal evolution of rural settlements, thereby enriching the research paradigm of urban and rural planning. Moreover, the multiple linkage analysis used in transportation engineering is employed, and five types of settlement evolution simulation processes are proposed based on mathematical language description, which enriches the construction and dynamic simulation technologies of complex networks. Finally, the object of research is the villages within small watersheds under dual influences of farmland afforestation and urbanization, which extends the research scope of complex network theory. This study can preferably reveal the urban–rural relationships within small watersheds, which can offer guidance for the resilience development in small watershed villages and provide a basis for guiding the regional urban–rural coordination, village layout, resource allocation, as well as construction guidelines. The advantage of complex network-based resilience evaluation lies in the investigation of complex relationships and dynamic changes. Compared to the traditional research, it sacrifices the diversity of settlement attributes and inter-settlement connections. Therefore, further optimization is needed to concern the interpretation of settlement system by complex network.

Based on complex network, this study puts forward eight indicator types for resilience evaluation. Our future research will focus on the diversity of network edges and nodes and seeking region-specific resilience evaluation methods. Additionally, we will achieve resilience research of the social–ecological system by further introducing ecological network into the social network. Hence, two-mode and two-level networks are also our future research interests.

**Supplementary Materials:** The following are available online at https://www.mdpi.com/article/10
.3390/land10101068/s1.

**Author Contributions:** Conceptualization, J.Z.; formal analysis, J.Z.; investigation, J.Z.; project administration, Q.H.; supervision, Q.H.; writing—original draft preparation, J.Z.; writing—review and
editing, Q.H. and J.Z. All authors have read and agreed to the published version of the manuscript.

**Funding:** This work was funded by the National Natural Science Foundation of China under Grant
51978058, National Natural Science Foundation of China under Grant 52178030, Major Theoretical
and Practical Research Project in the Social Sciences in Shaanxi under 2021ND0458. Fundamental
Research Funds for the Central Universities, CHD 300102411608.

**Conflicts of Interest:** The authors declare no conflict of interest.

**Appendix A**

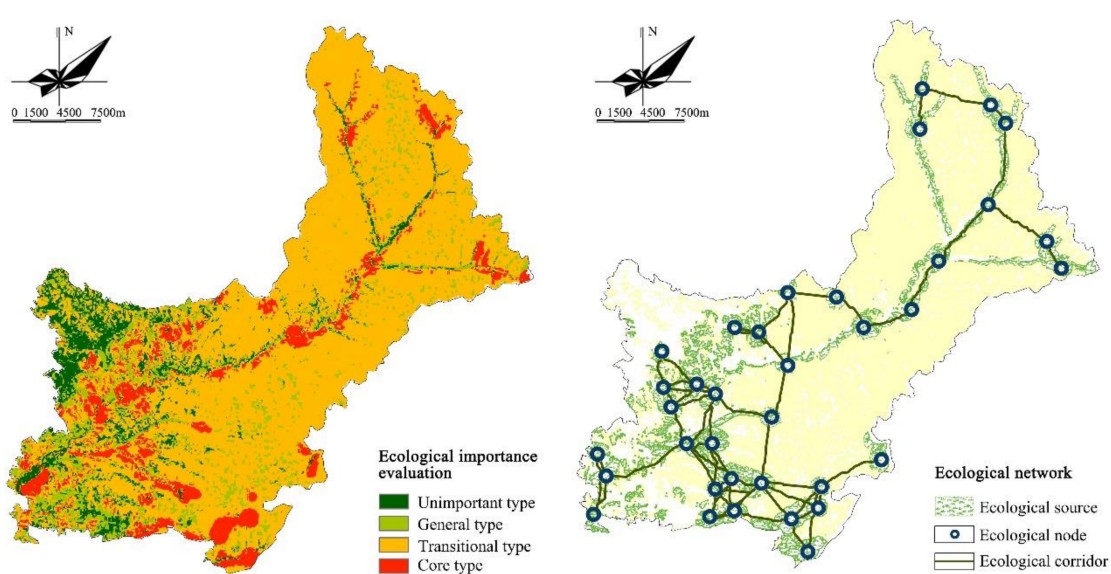

**Figure A1.** Ecological network construction results for the Sanshui Watershed.

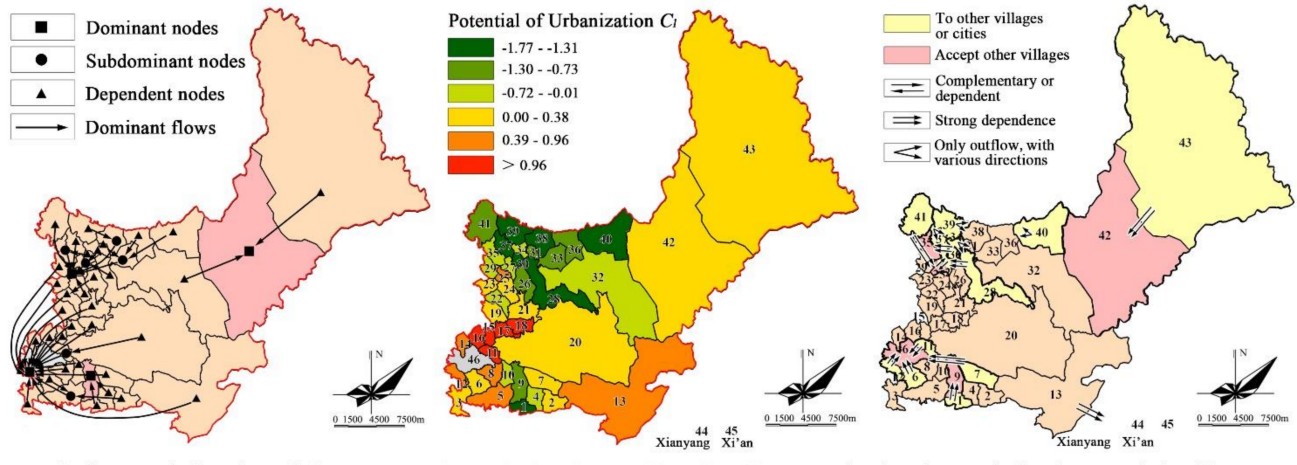

**Figure A2.** Population mobility, urban impact and relationship change trends of the study area.

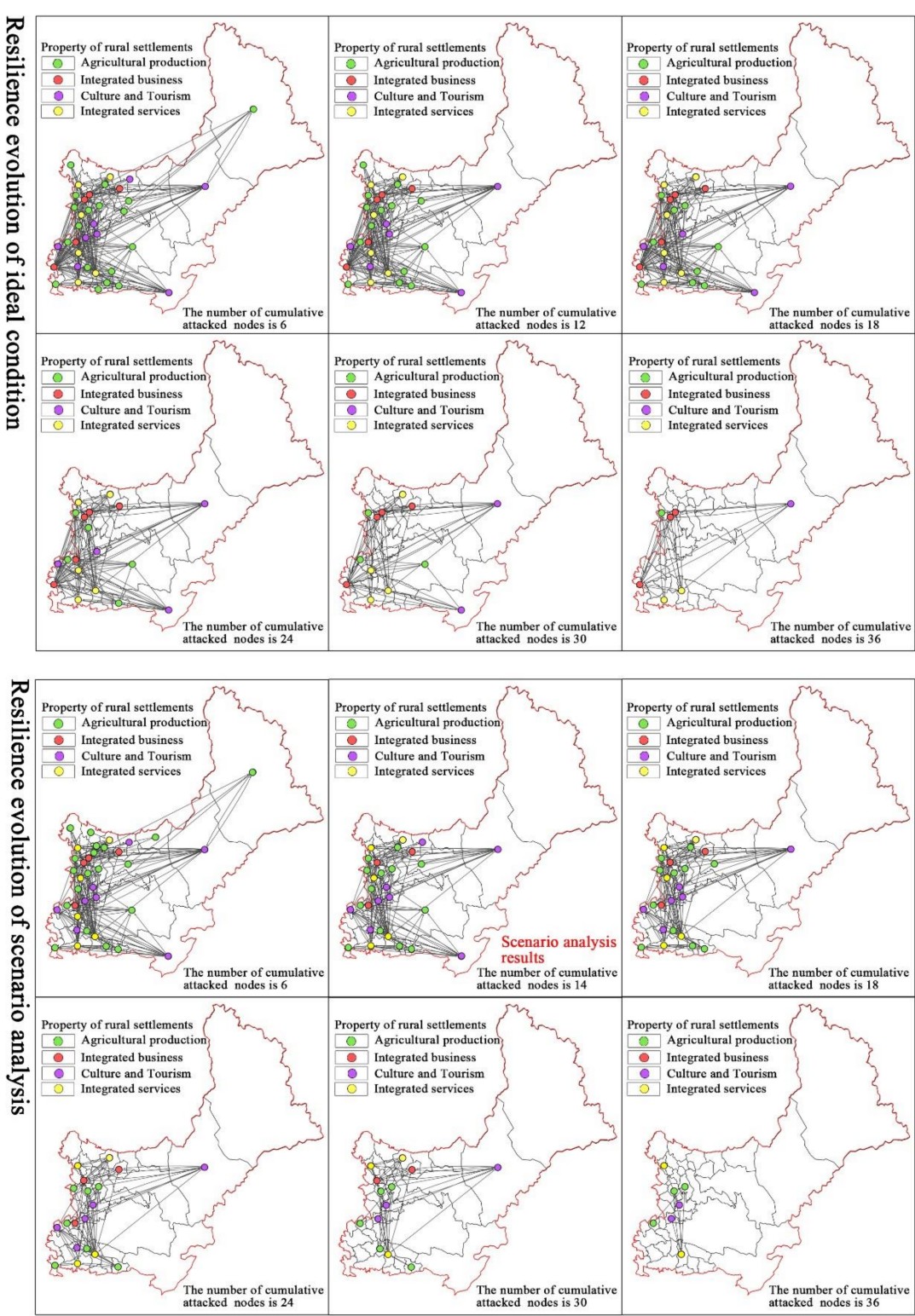

**Figure A3.** Comparisons of space evolution under the ideal condition versus scenario analysis.

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
