# Peer review of "Complex Network-Based Research on the Resilience of Rural Settlements in Sanshui Watershed"

_land, doi:10.3390/land10101068_

Round 1

Reviewer 1 Report

This paper takes the Sanshui River basin as an example to study the rural settlement system based on complex networks, and establishes a research framework of "spatial simulation-elastic evaluation-spatial planning". It has good innovation and theoretical value. The research results include the evolution trend of rural space from now to the future, as well as its static and dynamic spatial elasticity. The research method of this research is not mature enough and needs to be continuously optimized, especially in the selection of the resilience evaluation index system. The research results are of little significance, because this research is a spatial simulation of idealized conditions. How big is the difference in the real world? Debatable. Overall, I would like to recommend its publication after some revisions.

1.Problem with writing

(1)The word order is disordered and repetitive. Two sentences repeatedly describe that the small watershed village is a complete, dynamic and complex system.

The rural settlements within the watershed, which are connected closely and affected by the regional central cities, manifest integrity, complexity and dynamics.  (1. Introduction, line 39)

Accordingly, small watershed villages and surrounding regions/cities form an integrated, dynamic and complex system,(1. Introduction, Line 51)

(2) The logical sequence of writing. The introduction of the research object in the introduction part can be placed in the research data and methods part and then described in detail.

Villages in the Sanshui Watershed, the object of this study, are located within Xunyi County in the hilly–gully Loess Plateau, which are under jurisdiction of Xianyang and have certain connections with Xi'an,It covers an area of 1203.3 square kilometers,

(1. Introduction, Line 60)

(3) The graphic representation is not standardized. Two figures 1 and two figures 2 appear.

The latter category involves the interdependency variation trends of social–ecological networks, which determine the variation results of settlement nature (Fig. 1) (3.2.1 Topological simulation of settlement space, line 208)

Among them, the ecological network is constructed by consulting Allen and related mature approaches, which are not detailed herein. The results are shown in the Fig. S1 .(3.2.1 Topological simulation of settlement space, line 210)

which covers three types of conditions: scenario analysis, random conditions and extreme conditions (Fig. 2).(3.2.2 Settlement space change simulation, line 221)

The results of population mobility, urban impact and relationship change trends are shown in Fig. S2,(4.1 Space simulation results, line 402)

(4) Wrong expression. A village node with an unknown serial number appears.

Advantageous administrative villages, including future nodes 30, 6, 93 and 78, exhibit a weakened regional controllability, while those closer to the center, such as 61, 123, 129 and 102, show an enhanced controllability.(4.2 Static Resilience Assessment, Line 435)

(5)Discussion and conclusion expression issues. Take general results as conclusions, and lack a summary of the results.

In the future, the trend of settlement centralization will become more prominent, and the regional controllability of such advantageous villages as nodes 5 and 33 will be weakened, whose status will gradually be replaced by village 18.(6. Conclusion, line 691)

(6) The overall content expression problem. The full text needs to be more clear and concise in the expression of part of the content on the complex network, so as to facilitate the reading of readers outside the professional field.

2. Methodological issues

(1) Data selection problem There are many types of mobile phone signaling receipts. What is the relationship between different types of data and the research model?

Among them, the cellphone signaling data is purchased from Intelligent Footprint company, which comes from the active and passive records of interactions between the user cellphones and the base stations, including such behaviors as powering on and off, making calls, sending and receiving SMSs, surfing the Internet, periodic location updating, location area switching and 3G–to–4G conversion.(3.1.1 Research data ,line 128)

(2) Data processing issues How to process various types of data after obtaining the data, the article does not explain in detail. (3.1.2 Data preprocessing line 139)

(3) Model building problem How to write the processed data into the space model for simulation? What attributes does the information of each space node include?

the node and edge information in the present and future simulations of rural settlement space in the Sanshui Watershed is imported into Pajek software for modeling and analysis, and is visually represented in ArcGIS platform.(3.2.1 Topological simulation of settlement space, line 216)

(4) There are three kinds of change simulations in the change simulation, how to judge the difference between the real world and the change simulation. The three kinds of change simulations are all carried out by deleting the nodes or edges of the complex network. Are there any incremental and decremental changes in the elements contained in the nodes and edges?

which can be implemented by randomly deleting the nodes in network and the edges connected to them on the basis of present simulation network,(3.2.2 Simulation of changes in settlement space, line 223)

(5) How to determine static and dynamic indicators in the selection of resilience evaluation indicators? What is the measurement of the indicator after the indicator is determined? How to get the node degree? How to obtain structural holes? (3.3 Resilience Evaluation ,Line 251)

(6) How are the types of villages classified in spatial planning? Are there any specific classification criteria? (3.4.2 Rural development type planning, line 366)

(7) It is mentioned in the space simulation results that 14 nodes will disappear in the future. What index does the article use to determine the disappearance of the node, and is there a specific measurement?

In the future, 14 nodes will disappear, which will be relocated towards other villages (cities).(4.1 Space simulation result, line 407)

(8) In the static evaluation, a lot of analysis has been done on the five indicators of the static evaluation, but they all focus on a single indicator. Can we summarize the resilience of each node in the static evaluation through these analyses? (4.2 Static resilience evaluations, line 413)

(9) In the spatial planning results, how to determine which type of development direction each node belongs to, and need to be quantified, what is the specific measurement? Including the measurement of the central village. (4.4 Spatial planning result, line 575)

(10) In the final result, can we summarize the characteristics of the nodes with stronger resilience through a large number of analyses, and provide suggestions for future rural development. (6. Conclusion ,Line 680)

Author Response

Dear Sir or Madam

It's my pleasure and honour to come across a reviewer like you. Your suggestion is very detailed, which played a great role in enriching my paper and my work. Thank you very much for your advice. I agree with your comments on my article totally and sincerely, and I have tried my best to revise my paper that follows your evaluation. The following is my reply to you:

Reviewer 2 Report

The manuscript is adequately and good presented, the methods and results are clearly described. In general the thematic is significance. However, the maps could have been presented in a better way.

Author Response

Dear Sir or Madam

Thank you for reviewing my paper and giving a good evaluation. The following is my reply to you:

Reviewer 3 Report

The paper is focused on an important topic and shows several elaborations. It has good potential, however, in my opinion, several elements need to be improved. The paper is difficult to read as some methodological passages are unclear. I suggest that the following aspects be considered:

In the analysis of the literature (lines 70-80) it seems that the choice of network-based research method is justified by standing that the resilience index-based evaluation method have several difficulties applied in rural areas that are complex and adaptive territorial systems where the sum of the individual components is not always equal to the total. However this rule it is true for any complex dynamic system, whether urban or rural, so the preferences in the choice of method are not very clear.

In the description of the method (lines 148-150) it is indicated that the natural setting was first determined through the development of a series of indices; what are these indices? and how were they chosen?

The research mentions a questionnaire (lines 134-137) submitted to the administrations, but: what questions were asked? and how relevant were the answers in defining the resilience of rural systems?

The research mentions social-ecological networks several times, but it is not well understood how the social and ecological networks interact with each other; what are the variables associated with the nodes of the ecological network and the links that connect them? what influence they suffer from the social network? and how is the impact measured? 

Still on the subject of the ecological network, on the last page a diagram of the ecological network is proposed, but the document does not indicate anything about its operation. It indicates categories that classify the levels of protection and importance within the network without indicating the criteria and methodology for identifying these areas.

Author Response

Dear Sir or Madam

It's my pleasure and honour to come across a reviewer like you. Your suggestion is very detailed, which played a great role in enriching my paper and my work. Thank you very much for your advice. I agree with your comments on my article totally and sincerely, and I have tried my best to revise my paper that follows your evaluation. However, this study interprets complex systems based on systematic thinking, which involved complex methods and contents relatively. It could not present too much research process in a paper. I'm sorry for the difficult reading, and hope this modification could meet the requirements. The following is my reply to you:

Reviewer 4 Report

I was very pleased to read the article "Complex Network-based Research on the Resilience of Rural Settlements in Sanshui Watershed". The article submitted to me for review is properly written, contains all the elements required in a scientific publication, the research part is widely presented, the results are presented comprehensively and clearly, richly illustrated. The article is in line with current research trends.

I have a few minor comments that do not lower the high rating of the article.

  1. lines 115-123 do not fit into the Literature Review chapter. I suggest moving them to either the Introduction or Methods chapters.
  2. Reading the research results and their discussion, it seems that the purpose of the research written in lines 66-68 should be more elaborated.
  3. I miss in the Literature Review chapter a wider reference to the international scientific literature on the theory of networks.
  4. Please complete the figures showing the analysis area with the North direction and scale.
  5. I don't quite understand the context of farmland afforestation (line 7). I don't see any reference to it in the research done?

After making minor corrections and additions, the article is suitable for publication.      

Author Response

(The authors gave the same response as above.)

Round 2

Reviewer 3 Report

I have read the authors' replies and the revised document. I believe that the amendments and supplements meet the requirements. Authors have adequately implemented the paper.